# Metric Based Few-Shot Graph Classification

**Donato Crisostomi**[1]   **Simone Antonelli**[1 2]   **Valentino Maiorca**[1]   **Luca Moschella**[1]
**Riccardo Marin**[1 3]   **Emanuele Rodolà**[1]

[1]Sapienza University of Rome      [2]CISPA Helmholtz Center for Information Security
[3]University of Tübingen, Tübingen AI Center

{crisostomi, maiorca, moschella, rodola}@di.uniroma1.it
riccardo.marin@mnf.uni-tuebingen.de
simone.antonelli@cispa.de

## Abstract

Few-shot graph classification is a novel yet promising emerging research field that still lacks the soundness of well-established research domains. Existing works often consider different benchmarks and evaluation settings, hindering comparison and, therefore, scientific progress. In this work, we start by providing an extensive overview of the possible approaches to solving the task, comparing the current state-of-the-art and baselines via a unified evaluation framework. Our findings show that while graph-tailored approaches have a clear edge on some distributions, easily adapted few-shot learning methods generally perform better. In fact, we show that it is sufficient to equip a simple metric learning baseline with a state-of-the-art graph embedder to obtain the best overall results. We then show that straightforward additions at the latent level lead to substantial improvements by introducing i) a task-conditioned embedding space ii) a MixUp-based data augmentation technique. Finally, we release a highly reusable codebase to foster research in the field, offering modular and extensible implementations of all the relevant techniques.

## 1 Introduction

Graphs have ruled digital representations since the dawn of computer science. Their structure is simple and general, and their structural properties are well studied. Given the success of deep learning in different domains that enjoy a regular structure, such as those found in computer vision [4, 48, 72] and natural language processing [9, 14, 39, 55], a recent line of research has sought to extend it to manifolds and graph-structured data [3, 8, 26]. Nevertheless, the expressivity brought by deep learning comes at a cost: deep models require vast amounts of data to search the complex hypothesis spaces they define. When data is scarce, these models end up overfitting the training set, hindering their generalization capability on unseen samples. While annotations are usually abundant in computer vision and natural language processing, they are harder to obtain for graph-structured data due to the impossibility or expensiveness of the annotation process [29, 50, 52]. This is particularly true when the samples come from specialized domains such as biology, chemistry and medicine [28], where graph-structured data are ubiquitous. The most heartfelt example is drug testing, requiring expensive in-vivo testing and laborious wet experiments to label drugs and protein graphs [37].

To address this problem, the field of Few-Shot Learning [18, 20] aims at designing models which can effectively operate in scarce data scenarios. While this well-established research area enjoys a plethora of mature techniques, robust benchmarks and libraries, its intersection with graph representation learning is still at an embryonic stage. As such, the field suffers from a lack of uniformity: existing works often consider different benchmarks and evaluation settings, with no two works considering the same set of datasets or evaluation hyperparameters. This scenario results in a fragmented understanding, hindering comparison and, therefore, scientific progress in the field. In an attempt to

D. Crisostomi et al., Metric Based Few-Shot Graph Classification. *Proceedings of the First Learning on Graphs Conference (LoG 2022)*, PMLR 198, Virtual Event, December 9–12, 2022.

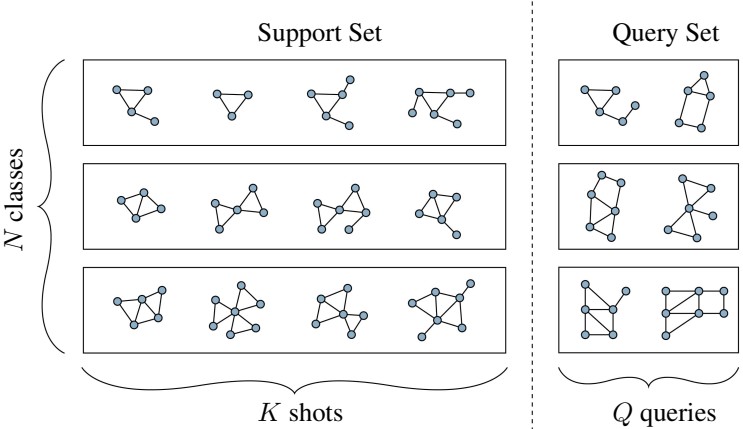

**Figure 1:** An $N$-way $K$-shot episode. In this example, there are $N = 3$ classes. Each class has $K = 4$ supports yielding a support set with size $N * K = 12$. The class information provided by the supports is exploited to classify the queries. We test the classification accuracy on all $N$ classes. In this figure there are $Q = 2$ queries for each class, thus the query set has size $N * Q = 6$.

mitigate this issue and facilitate new research, we provide a modular and easily extensible codebase[1] with re-implementations of the most relevant baselines and state-of-the-art works. The latter allows both for straightforward use by practitioners and for a fair comparison of the techniques in a unified evaluation setting. Our findings show that kernel methods achieve impressive results on particular distributions but are too rigid to be used as an overall solution. On the other hand, few-shot learning techniques can be easily adapted to the graph setting by employing a graph neural network as an encoder. We argue that the latter is sufficient to capture the complexity of the structure, relieving the remaining pipeline of the burden. When in the latent space, standard techniques behave as expected and no further tailoring to the graph domain is needed.

In this direction, we show that a simple Prototypical Network [49] architecture outperforms existing works when equipped with a state-of-the-art graph embedder. As typical in few-shot learning, we frame tasks as episodes, where an episode is defined by a set of classes and several supervised samples (supports) for each of them [57]. Such an episode is depicted in Figure 1. This setting favors a straightforward addition to the architecture: in fact, while a standard Prototypical Network would embed the samples in the same way independently of the episode, we draw inspiration from [40] and empower the graph embeddings by conditioning them on the particular set of classes seen in the episode. This way, the intermediate features and the final embeddings may be modulated according to what is best for the current episode. Finally, we propose to augment the training dataset using a MixUp-based [71] online data augmentation technique. The latter creates artificial samples from two existing ones as a mix-up of their latent representations, probing unexplored regions of the latent space that can accommodate samples from unseen classes. We finally show that these additions are beneficial for the task, both qualitatively and quantitatively.

Summarizing, our contribution is 4-fold:

1. We provide an extensive overview of the possible approaches to solve the few-shot classification task, comparing all the existing works and baselines in a unified evaluation framework;

2. We release a strongly re-usable codebase to foster research in the field, offering modular and extensible implementations of all the relevant techniques;

3. We show that it is sufficient to equip existing few-shot pipelines with graph encoders to obtain competitive results, proposing in particular a metric learning baseline for the task;

4. We equip the latter with two supplementary modules: an episode-adaptive embedder and a novel online data augmentation technique, showing their benefits qualitatively and quantitatively.

---

[1]https://github.com/crisostomi/metric-few-shot-graph

## 2 Related work

**Few-Shot Learning.** Data-scarce tasks are usually tackled by using one of the following paradigms: i) *transfer learning* techniques [1, 34, 35] that aim at transferring the knowledge gained from a data-abundant task to a task with scarce data; ii) *meta-learning* [21, 42, 70] techniques that more generally introduce a meta-learning procedure to gradually learn meta-knowledge that generalizes across several tasks; iii) *data augmentation* works [22, 54, 66] that seek to augment the data applying transformations on the available samples to generate new ones preserving specific properties. We refer the reader to [62] for an extensive treatment of the matter. Particularly relevant to our work are metric learning approaches. In this area, [57] suggest embedding both supports and queries and then labeling the query with the label of its nearest neighbor in the embedding space. By obtaining a class distribution for the query using a softmax over the distances from the supports, they then learn the embedding space by minimizing the negative log-likelihood. [49] generalize this intuition by allowing $K$ supports for class to be aggregated to form prototypes. Given its effectiveness and simplicity, we chose this approach as the starting point for our architecture.

**Graph Data Augmentation.** Data augmentation follows the idea that in the working domain, there exist transformations that can be applied to samples to generate new ones in a controlled way (*e.g.*, preserving the sample class in a classification setting while changing its content). Therefore, synthetic samples can meet the needs of large neural networks that require training with high volumes of data [62]. In Euclidean domains (*e.g.*, images), this can often be achieved by simple rotations and translations [5, 43]. Unfortunately, in the graph domain, it is challenging to define such transformations on a given graph sample while keeping control of its properties. To this end, a line of works takes inspiration from Mix-Up [38, 71] to create new artificial samples as a combination of two existing ones: [24, 27, 41, 64] propose to augment graph data directly in the data space, while [65] interpolates latent representations to create novel ones. We also operate in the latent space, but differently from [65], we suggest creating a new sample by selecting only certain features of one representation and the remaining ones from the other by employing a random gating vector. This allows for obtaining synthetic samples as random compositions of the features of the existing samples, rather than a linear interpolation of them. We also argue that the proposed Mix-Up is tailored for metric learning, making full use of the similarity among samples and class prototypes.

**Few-Shot Graph Representation Learning.** Few-shot graph representation learning is concerned with applying graph representation learning techniques in scarce data scenarios. Similarly to standard graph representation learning, it tackles tasks at different levels of granularity: node-level [15, 59, 69, 73, 74], edge-level [2, 36, 44, 60], and graph-level [12, 25, 30, 33, 37, 61, 63]. Concerning the latter, `GSM` [12] proposes a hierarchical approach, `AS-MAML` adapts the well known `MAML` [21] architecture to the graph setting, and `SMF-GIN` [30] uses a Prototypical Network (PN) variant with domain-specific priors. Differently from the latter, we employ a more faithful formulation of PN that shows far superior performance. This difference is further discussed in Appendix B.4. Most recently, `FAITH` [61] proposes to capture episode correlations with an inter-episode hierarchical graph, while `SP-NP` [33] suggests employing neural processes [23] for the task.

## 3 Approach

**Setting and Notation.** In few-shot graph classification each sample is a tuple $(\mathcal{G} = (\mathcal{V}, \mathcal{E}), y)$ where $\mathcal{G} = (\mathcal{V}, \mathcal{E})$ is a graph with node set $\mathcal{V}$ and edge set $\mathcal{E}$, while $y$ is a graph-level class. Given a set of data-abundant *base* classes $C_b$, we aim to classify a set of data-scarce *novel* classes $C_n$. We cast this problem through an episodic framework [58]; during training, we mimic the few-shot setting by dividing the base training data into episodes. Each episode $e$ is a $N$-way $K$-shot classification task, with its own train ($D_{train}$) and test ($D_{test}$) data. For each of the $N$ classes, $D_{train}$ contains $K$ corresponding *support* graphs, while $D_{test}$ contains $Q$ *query* graphs. A schematic visualization of an episode is depicted in Figure 1. We refer the reader to Appendix B.2 for an algorithmic description of the episode generation.

**Prototypical Network (PN) Architecture.** We build our network upon the simple-yet-effective idea of Prototypical Networks [49], originally proposed for few-shot image classification. We employ a state-of-the-art Graph Neural Network as node embedder, composed of a set of layers of GIN convolutions [68], each equipped with a MLP regularized with GraphNorm [10]. In practice, each

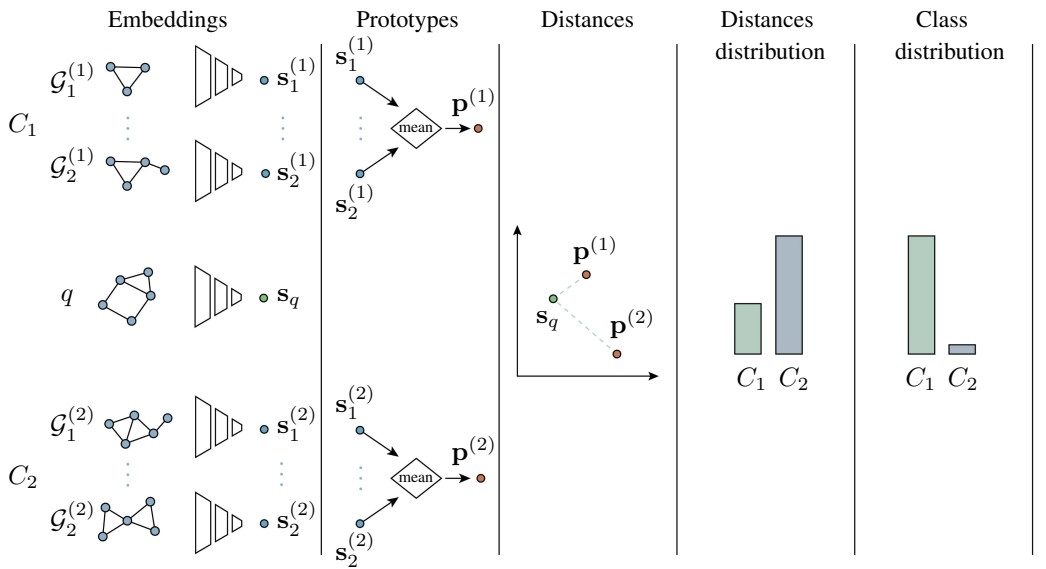

**Figure 2:** Prototypical Networks architecture. A graph encoder embeds the supports graphs, the embeddings that belong to the same class are averaged to obtain the class prototype $p$. To classify a query graph $q$, it is embedded in the same space of the supports. The distances in the latent space between the query and the prototypes determine the similarities and thus the probability distribution of the query among the different classes, computed as in Equation (3).

sample is first passed through a set of convolutions, obtaining a hidden representation $\mathbf{h}^{(\ell)}$ for each layer. According to [68], the latter is obtained by updating at each layer its hidden representation as

$$\mathbf{h}_v^{(\ell)} = \mathrm{MLP}^{(\ell)}\left(\left(1 + \epsilon^{(\ell)}\right) \cdot \mathbf{h}_v^{(\ell-1)} + \sum_{u \in \mathcal{N}(v)} \mathbf{h}_u^{(\ell-1)}\right), \tag{1}$$

where $\epsilon^{(\ell)}$ is a learnable parameter. Following [67], the final node $d$-dimensional embedding $\mathbf{h}_v \in R^d$ is then given by the concatenation of the outputs of all the layers. The graph-level embedding is then obtained by employing a global pooling function, such as mean or sum. While the sum is a more expressive pooling function for GNNs [68], we observed the mean to behave better for the task and will therefore be adopted when not specified differently. The $K$ embedded supports $\mathbf{s}_1^{(n)}, \ldots, \mathbf{s}_K^{(n)}$ for each class $n$ are then aggregated to form the class prototypes $\mathbf{p}^{(n)}$,

$$\mathbf{p}^{(n)} = \frac{1}{K} \sum_{k=1}^{K} \mathbf{s}_k^{(n)}. \tag{2}$$

Similarly, the $Q$ query graphs for each class $n$ are embedded to obtain $\mathbf{q}_1^{(n)}, \ldots, \mathbf{q}_Q^{(n)}$. To compare each query graph embedding $\mathbf{q}$ with the class prototypes $\mathbf{p}_1, \ldots, \mathbf{p}_N$, we use the $L_2$ metric scaled by a learnable temperature factor $\alpha$ as suggested in [40]. We refer to this metric as $d_\alpha$. The class probability distribution $\boldsymbol{\rho}$ for the query is finally computed by taking the softmax over these distances:

$$\boldsymbol{\rho}_n = \frac{\exp\left(-d_\alpha(\mathbf{q}, \mathbf{p}_n)\right)}{\sum_{n'=1}^{N} \exp(-d_\alpha(\mathbf{q}, \mathbf{p}_{n'}))}. \tag{3}$$

The model is then trained end-to-end by minimizing via SGD the log-probability $\mathcal{L}(\phi) = -\log \boldsymbol{\rho}_n$ of the true class $n$. We will refer to this approach without additions as PN in the experiments.

**Task-Adaptive Embedding (TAE).** Until now, our module computes the embeddings regardless of the specific composition of the episode. Our intuition is that the context in which a graph appears should affect its representation. In practice, inspired by [40], we condition the embeddings on the particular task (episode) for which they are computed. Such influence will be expressed by a translation $\boldsymbol{\beta}$ and a scaling $\boldsymbol{\gamma}$.

First of all, given an episode $e$ we compute an episode representation $\mathbf{p}_e$ as the mean of the prototypes $\mathbf{p}_n$ for the classes $n = 1, \ldots, N$ in the episode. We consider $\mathbf{p}_e$ as a prototype for the episode and a

proxy for the task. Then, we feed it to a *Task Embedding Network* (TEN), composed of two distinct residual MLPs. These output a shift vector $\boldsymbol{\beta}^{(\ell)}$ and a scale vector $\boldsymbol{\gamma}^{(\ell)}$ respectively for each layer of the graph embedding module. At layer $\ell$, the output $\mathbf{h}^{(\ell)}$ is then conditioned on the episode by transforming it as

$$\hat{\mathbf{h}}^{(\ell)} = \boldsymbol{\gamma} \odot \mathbf{h}^{(\ell)} + \boldsymbol{\beta}\,. \tag{4}$$

As in [40], at each layer $\boldsymbol{\gamma}$ and $\boldsymbol{\beta}$ are multiplied by two $L_2$-penalized scalars $\gamma_0$ and $\beta_0$ so as to promote significant conditioning only if useful. Wrapping up, defining $g_\Theta$ and $h_\Phi$ to be the predictors for the shift and scale vectors respectively, the actual vectors to be multiplied by the hidden representation are respectively $\boldsymbol{\beta} = \beta_0 g_\Theta(\mathbf{p}_e)$ and $\boldsymbol{\gamma} = \gamma_0 h_\Phi(\mathbf{p}_e) + \mathbf{1}$. When we use this improvement in our experiments, we add the label `TAE` to the method name.

**MixUp (MU) Embedding Augmentation.** Typical learning pipelines rely on data augmentation to overcome limited variability in the dataset. While this is mainly performed to obtain invariance to specific transformations, we use it to improve our embedding representation, promoting generalization on unseen feature combinations. In practice, given an episode $e$, we randomly sample for each pair of classes $n_1, n_2$ two graphs $\mathcal{G}^{(1)}$ and $\mathcal{G}^{(2)}$ from the corresponding support sets. Then, we compute their embeddings $\mathbf{s}^{(1)}$ and $\mathbf{s}^{(2)}$, as well as their class probability distributions $\boldsymbol{\rho}^{(1)}$ and $\boldsymbol{\rho}^{(2)}$ according to Equation (3). Next, we randomly obtain a boolean mask $\boldsymbol{\sigma} \in \{0, 1\}^d$. We can then obtain a novel synthetic example by mixing the features of the two graphs in the latent space:

$$\tilde{\boldsymbol{s}} = \boldsymbol{\sigma} \odot \mathbf{s}^{(1)} + (\mathbf{1} - \boldsymbol{\sigma}) \odot \mathbf{s}^{(2)}\,, \tag{5}$$

where $\mathbf{1}$ is a $d$-dimensional vector of ones and $\odot$ denotes component-wise product. Finally, we craft a synthetic class probability $\tilde{\boldsymbol{\rho}}$ for this example by linear interpolation:

$$\tilde{\boldsymbol{\rho}} = \lambda \boldsymbol{\rho}^{(1)} + (1 - \lambda)\boldsymbol{\rho}^{(2)}, \quad \lambda = \left(\frac{1}{d}\sum_{i=1}^{d}\boldsymbol{\sigma}_i\right) \tag{6}$$

where $\lambda$ represents the percentage of features sampled from the first sample. If we then compute the class distribution $\boldsymbol{\rho}$ for $\tilde{\mathbf{s}}$ according to Equation (3), we can require it to be similar to Equation (6) by adding the following regularizing term to the training loss:

$$\mathcal{L}_{\mathrm{MU}} = \|\boldsymbol{\rho} - \tilde{\boldsymbol{\rho}}\|_2^2\,. \tag{7}$$

Intuitively, by adopting this online data augmentation procedure, the network is faced with new feature combinations during training, helping to explore unseen regions of the embedding space. Moreover, we argue that in a metric learning approach, the distances with respect to all the prototypes should be considered, and not only the ones corresponding to the classes that are used for interpolation. On the other hand, in standard MixUp [71], the label for the new artificial sample $x' = \alpha x_1 + (1 - \alpha x_2)$ is obtained as the linear interpolation of the one-hot ground-truth vectors $y_1$ and $y_2$. This way, the information only considers the distance/similarity w.r.t. the classes of the two original samples. On the contrary, the proposed augmentation also maintains information on the distance from all the other prototypes and hence classes, thereby providing finer granularity than mixing one-hot ground truth vectors. The overall procedure is summarized in Figure 3.

## 4 Experiments

### 4.1 Datasets

We benchmark our approach over two sets of datasets: the first one was introduced in [12], and consists of: (i) `TRIANGLES`, a collection of graphs labeled $i = 1, \ldots, 10$, where $i$ is the number of triangles in the graph. (ii) `ENZYMES`, a dataset of tertiary protein structures from the BRENDA database [11]; each label corresponds to a different top-level enzyme. (iii) `Letter-High`, a collection of graph-represented letter drawings from the English alphabet; each drawing is labeled with the corresponding letter. (iv) `Reddit-12K`, a social network dataset where graphs represent threads, with edges connecting users interacting. The corresponding discussion forum gives the label of a thread. We will refer to this set of datasets as $\mathcal{D}_\mathrm{A}$. The second set of datasets was introduced in [37] and consists of: (i) `Graph-R52`, a textual dataset in which each graph represents a different text, with words being connected by an edge if they appear together in a sliding window. (ii) `COIL-DEL`, a

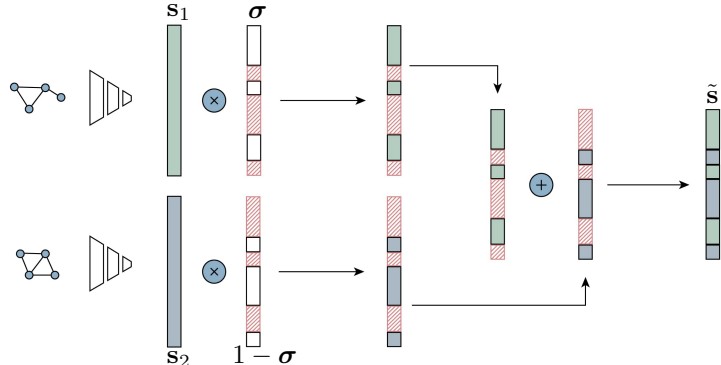

**Figure 3:** Mixup procedure. Each graph is embedded into a latent representation. We generate a random boolean mask $\sigma$ and its complementary $1 - \sigma$, which describe the features to select from $s_1$ and $s_2$. The selected features are then recomposed to generated the novel latent vector $\tilde{s}$.

| | Model | TRIANGLES | | Letter-High | | ENZYMES | | Reddit | | mean | |
|---|---|---|---|---|---|---|---|---|---|---|---|
| | | 5-shot | 10-shot | 5-shot | 10-shot | 5-shot | 10-shot | 5-shot | 10-shot | 5-shot | 10-shot |
| Kernel | WL | $59.3 \pm 7.7$ | $64.5 \pm 7.4$ | $69.8 \pm 7.2$ | $74.1 \pm 5.8$ | $54.9 \pm 9.1$ | $57.0 \pm 9.1$ | $29.3 \pm 4.5$ | $34.2 \pm 4.9$ | 53.3 | 57.5 |
| | SP | $61.0 \pm 8.0$ | $66.7 \pm 7.4$ | $67.3 \pm 6.8$ | $71.2 \pm 6.6$ | $58.8 \pm 9.1$ | $61.5 \pm 8.8$ | $51.0 \pm 5.8$ | $52.7 \pm 4.9$ | 59.5 | 63.0 |
| | Graphlet | $69.2 \pm 10.2$ | $79.3 \pm 8.1$ | $35.4 \pm 4.2$ | $39.4 \pm 4.4$ | $\mathbf{58.8} \pm 10.6$ | $59.8 \pm 9.8$ | $42.7 \pm 11.3$ | $45.4 \pm 11.2$ | 51.5 | 56.0 |
| Meta | MAML | $\mathbf{87.8} \pm 4.9$ | $\mathbf{88.2} \pm 4.5$ | $69.6 \pm 7.9$ | $73.8 \pm 5.7$ | $52.7 \pm 8.9$ | $54.9 \pm 8.5$ | $26.0 \pm 6.0$ | $37.0 \pm 6.9$ | 59.0 | 63.5 |
| | AS-MAML [37] | $86.4 \pm 0.7$ | $87.2 \pm 0.6$ | $76.2 \pm 0.8$ | $77.8 \pm 0.7$ | - | - | - | - | - | - |
| | AS-MAML* | $79.2 \pm 5.9$ | $84.0 \pm 5.3$ | $71.8 \pm 7.6$ | $73.0 \pm 5.2$ | $45.1 \pm 8.2$ | $53.1 \pm 8.1$ | $33.7 \pm 10.8$ | $37.4 \pm 10.8$ | 57.4 | 61.9 |
| Metric | SMF-GIN [30] | $79.8 \pm 0.7$ | - | - | - | - | - | - | - | - | - |
| | FAITH [61] | $79.5 \pm 4.0$ | $80.7 \pm 3.5$ | $71.5 \pm 3.5$ | $76.6 \pm 3.2$ | $57.8 \pm 4.6$ | $\mathbf{62.1} \pm 4.1$ | $42.7 \pm 4.1$ | $46.6 \pm 4.0$ | 62.9 | 66.5 |
| | SPNP [33] | $85.2 \pm 0.7$ | $86.8 \pm 0.7$ | - | - | - | - | - | - | - | - |
| Transfer | GIN | $82.1 \pm 6.3$ | $83.6 \pm 5.4$ | $68.4 \pm 7.3$ | $74.5 \pm 5.7$ | $54.2 \pm 9.3$ | $55.9 \pm 9.4$ | $49.8 \pm 7.0$ | $\mathbf{53.4} \pm 6.3$ | 63.6 | 66.8 |
| | GAT | $82.8 \pm 6.1$ | $83.4 \pm 5.5$ | $74.1 \pm 6.2$ | $76.4 \pm 5.1$ | $53.6 \pm 9.4$ | $55.4 \pm 9.1$ | $39.0 \pm 6.7$ | $41.7 \pm 6.1$ | 62.4 | 64.2 |
| | GCN | $82.0 \pm 6.1$ | $82.7 \pm 5.5$ | $71.3 \pm 6.8$ | $74.9 \pm 5.5$ | $53.4 \pm 9.3$ | $54.6 \pm 9.4$ | $44.7 \pm 7.4$ | $50.8 \pm 6.3$ | 62.8 | 65.7 |
| | GSM [12] | $71.4 \pm 4.3$ | $75.6 \pm 3.6$ | $69.9 \pm 5.9$ | $73.2 \pm 3.4$ | $55.4 \pm 5.7$ | $60.6 \pm 3.8$ | $41.5 \pm 4.1$ | $45.6 \pm 3.6$ | 59.5 | 63.8 |
| | GSM* | $79.2 \pm 5.7$ | $81.0 \pm 5.6$ | $72.9 \pm 6.4$ | $75.6 \pm 5.6$ | $56.8 \pm 10.3$ | $58.4 \pm 9.7$ | $40.7 \pm 6.8$ | $46.4 \pm 6.3$ | 62.4 | 65.4 |
| Ours | PN+TAE+MU | $87.4 \pm 0.9 \pm 4e\text{-}4$ | $87.5 \pm 0.8 \pm 3e\text{-}4$ | $\mathbf{77.2} \pm 5.5 \pm 2e\text{-}3$ | $\mathbf{79.2} \pm 4.8 \pm 1e\text{-}3$ | $56.8 \pm 10.1 \pm 4e\text{-}3$ | $59.3 \pm 9.4 \pm 3e\text{-}3$ | $45.7 \pm 6.7 \pm 2e\text{-}3$ | $48.5 \pm 6.3 \pm 2e\text{-}3$ | **66.8** | **68.7** |

**Table 1:** Macro accuracy scores over different *k-shot* settings and architectures. They are partitioned into baselines (upper section) and our full architecture (lower section). The best scores are in bold. We report standard deviation values in blue and $0.9$ confidence intervals in orange. Cells filled with - indicate lack of results in the original works for the corresponding datasets.

collection of graph-represented images obtained through corner detection and Delaunay triangulation. We will refer to this set of datasets as $\mathcal{D}_B$. The overall dataset statistics are reported in Appendix A.

It is important to note that only the datasets in $\mathcal{D}_B$ have enough classes to permit a disjoint set of classes for validation. In contrast, a disjoint subset of the training samples is used as a validation set in the first four by existing works. We argue that this setting is critically unfit for few-shot learning, as the validation set does not make up for a good proxy for the actual testing environment since the classes are not novel. Moreover, the lack of a reliable validation set prevents the usage of early stopping, as there is no way to decide on a good stopping criterion for samples from unseen classes. We nevertheless report the outcomes of this evaluation setting for the sake of comparison.

## 4.2 Baselines

We group the considered approaches according to their category. We note, however, that the taxonomy is not strict, and some works may belong to more categories.

**Graph kernels.** Starting from graph kernel methods, we consider `Weisfeiler-Lehman` (WL) [46], `Shortest Path` (SP) [7] and `Graphlet` [45]. These well-known methods compute similarity scores between pairs of graphs, and can be understood as performing inner products between graphs. We refer the reader to [32] for a thorough treatment. In our implementation, an SVM is used as the head classifier for all the methods. More implementation details can be found in Appendix B.

**Meta learning.**    Regarding the meta-learning approaches, we consider both vanilla Model-Agnostic Meta-Learning (MAML) [21] and its graph-tailored variant AS-MAML [37]. The former employs a meta-learner trained by optimizing the sum of the losses from a set of downstream tasks, encouraging the learning of features that can be adapted with a small number of optimization steps. The latter builds upon MAML by integrating a reinforcement learning-based adaptive step controller to decide the number of inner optimization steps adaptively.

**Metric learning.**    For the metric based approaches, the considered works are SMF-GIN [30], FAITH [61] and SPNP [33]. In SMF-GIN, a GNN is employed to encode both global (via an attention over different GNN layer encodings) and local (via an attention over different substructure encodings) properties. We point out that they include a ProtoNet-based baseline. However, their implementation does not accurately follow the original one and, differently from us, leverages domain-specific prior knowledge. FAITH proposes to capture correlations among meta-training tasks via a hierarchical task graph to transfer meta-knowledge to the target task better. For each meta-training task, a set of additional ones is sampled according to its classes to build the hierarchical graph. Subsequently, the knowledge from the embeddings extracted by the hierarchical task graph is aggregated to classify the query graph samples. Finally, SPNP makes use of Neural Processes (NPs) by introducing an encoder capable of constructing stochastic processes considering the graph structure information extracted by a GNN and a prototypical decoder that provides a metric space where classification is performed.

**Transfer learning.**    Finally, transfer learning approaches include GSM [12] and three simple baselines built on top of varying GNN architectures, namely GIN [68], GAT [56] and GCN [31]. The latter follow the most standard fine-tuning procedure, *i.e.* training the embedder backbone over the base classes and fine-tuning the classifier head over the $K$ supports. In GSM, graph prototypes are computed as a first step and then clustered based on their spectral properties to create super-classes. These are then used to generate a super-graph which is employed to separate the novel graphs. The original work however does not follow an episodic framework, making the results not directly comparable. For this reason, we also re-implemented it to cast it in the episodic framework. We refer the reader to Appendix B for more details.

### 4.3   Experimental details

Our graph embedder is composed of two layers of GIN followed by a mean pooling layer, and the dimension of the resulting embeddings is set to $64$. Furthermore, both the latent mixup regularizer and the L2 regularizer of the task-adaptive embedding are weighted at $0.1$. The framework is trained with a batch size of 32 using Adam optimizer with a learning rate of $0.0001$. We implement our framework with Pytorch Lightning [17] using Pytorch Geometric [19], and WandB [6] to log the experiment results. The specific configurations of all our approaches are reported in Appendix B.

## 5   Results

We report in this section the results over the two sets of benchmark datasets $\mathcal{D}_A$, $\mathcal{D}_B$. Given the lack of homogeneity in the evaluation settings of previous works, we will report both the standard deviation of our results between different episodes and the $0.95$ confidence interval. Moreover, when possible, we provide the re-implementation of the methods, indicating them with a $\star$.

**Benchmark $\mathcal{D}_A$.**    As can be seen in Table 1, there is no one-fits-all approach for the considered datasets. In fact, the best results for each are obtained with approaches belonging to different categories, including graph kernels. However, the proposed approach obtains the best results if we consider the average performance for both $K = 5, 10$. In fact, considering previous published works, we obtain an overall margin of $+7.3\%$, $+4.9\%$ accuracy for $K = 5, 10$ compared to GSM [12], $+9.4\%$ and $+6.8\%$ compared to to AS-MAML$^\star$ [37], and $+3.9\%$, $+2.2\%$ with respect to FAITH [61]. However, we again stress the partial inadequacy of these datasets as a realistic evaluation tool, given the lack of a disjoint set of classes for the validation set. Interestingly, our re-implementation of GSM$^\star$ obtains slightly better results than the original over Reddit and Letter-High, a significant improvement over TRIANGLES and a comparable result over ENZYMES. The difference may be attributed to the difference in the evaluation setting, as the non-episodic framework employed in GSM does not have a fixed number of queries per class, and batches are sampled without episodes.

| Category | Model | Graph-R52 | | | | COIL-DEL | | | | mean | |
|---|---|---|---|---|---|---|---|---|---|---|---|
| | | 5-shot | | 10-shot | | 5-shot | | 10-shot | | 5-shot | 10-shot |
| Kernel | WL | **88.2** | ±10.9 | **91.4** | ±9.1 | 56.5 | ±12.7 | 64.0 | ±12.8 | 72.4 | 77.7 |
| | SP | 84.3 | ±11.3 | 88.9 | ±9.6 | 39.6 | ±9.6 | 45.5 | ±11.3 | 61.9 | 67.2 |
| | Graphlet | 57.4 | ±10.3 | 58.3 | ±10.1 | 57.6 | ±12.2 | 61.3 | ±11.5 | 57.5 | 59.8 |
| Meta | MAML | 64.9 | ±13.3 | 70.1 | ±12.7 | 76.7 | ±12.6 | 78.8 | ±11.5 | 70.8 | 74.4 |
| | AS-MAML [37] | 75.3 | ±1.1 | 78.3 | ±1.1 | 81.5 | ±1.3 | 84.7 | ±1.3 | 78.4 | 81.5 |
| | AS-MAML* | 72.3 | ±14.8 | 72.0 | ±15.5 | 77.2 | ±11.1 | 80.1 | ±9.9 | 74.7 | 76.0 |
| Transfer | GIN | 67.2 | ±13.9 | 66.4 | ±13.7 | 72.3 | ±11.4 | 74.0 | ±11.3 | 69.8 | 74.4 |
| | GAT | 75.2 | ±12.8 | 77.5 | ±12.4 | 79.3 | ±10.3 | 80.8 | ±9.9 | 77.2 | 79.1 |
| | GCN | 75.1 | ±13.0 | 74.1 | ±14.5 | 75.2 | ±11.4 | 77.1 | ±10.8 | 75.1 | 75.6 |
| | GSM* | 70.3 | ±15.7 | 71.6 | ±14.9 | 74.9 | ±11.4 | 79.2 | ±10.3 | 72.6 | 75.4 |
| Metric | SPNP [33] | - | | - | | 84.8 | ±1.6 | 87.3 | ±1.6 | - | - |
| Ours | PN | 73.1 | ±12.1 | 78.0 | ±10.6 | 85.5 | ±9.8 | 87.2 | ±9.3 | 79.3 | 82.6 |
| | PN+TAE | 77.9 | ±11.8 | 81.3 | ±10.6 | 86.4 | ±9.6 | 88.8 | ±8.5 | 82.1 | 85.0 |
| | PN+TAE+MU | 77.9 | ±11.8 ±3e-3 | 81.5 | ±10.4 ±4e-3 | **87.7** | ±9.2 ±4e-3 | **90.5** | ±7.7 ±3e-3 | **82.8** | **86.0** |

**Table 2:** Macro accuracy scores over different *k-shot* settings and architecture. The best scores are in bold. We report standard deviation values in blue and 0.9 confidence intervals in orange. Cells filled with - indicates lack of results in the original works for the corresponding datasets.

**Benchmark $\mathcal{D}_\mathbf{B}$.** Table 2 shows the results for the two datasets in the benchmark. Most surprisingly, graph kernels exhibit superior performance over R-52, outperforming all the considered deep learning models. It must be noted, however, that the latter is characterized by a very skewed sample distribution, with few classes accounting for most of the samples. In this regard, deep learning methods may end up overfitting the most frequent class, while graph kernel methods are less prone due to the smaller parameter volume and stronger inductive bias. Nevertheless, the latter also hinders their adaptivity to different distributions: we can see, in fact, how the same methods perform miserably on COIL-DEL. This can be observed by considering the mean results over both sets of datasets, in which graph kernels generally perform the worst. Compared to existing works, our approach obtains an average margin of +4.37% and +4.53% over AS-MAML [37] and +10.2%, +10.6% over GSM for $K = 5, 10$ respectively. Finally, the last three rows of Table 2 show the efficacy of the proposed improvements. Task-adaptive embedding (TAE) allows obtaining the most critical gain, yielding an average increment of +2.82% and +2.42% for the 5-shot and 10-shot cases, respectively. Then, the proposed online data augmentation technique (MU) allows obtaining an additional boost, especially on COIL-DEL. In fact, in the latter case, its addition yields a +0.65% and +1.72% improvement in accuracy for $K = 5, 10$. We speculate that the less marked benefit on Graph-R52 may in part be caused of its highly skewed class distribution, as discussed in Appendix C.4. Remarkably, a vanilla Prototypical Network (PN) architecture with the proposed graph embedder is already sufficient to obtain state-of-the-art results.

**Qualitative analysis.** The latent space learned by the graph embedder is the core element of our approach since it determines the prototypes and the subsequent sample classification. To provide a better insight into our method peculiarities, Figure 5 depicts a T-SNE representation of the learned embeddings for novel classes. Each row represents different episodes, while the different columns show the different embeddings obtained with our approach and its further refinements. We also highlight the queries (crosses), the supports (circles) and the prototypes (star). As can be seen, our approach separates samples belonging to novel classes into clearly defined clusters. Already in PN, some classes naturally cluster in different regions of the embedding. The TAE regularization improves the class separation without significantly changing the disposition of the clusters in the space. Our insight is that the context may let the network reorganize the already seen space without moving far from the already obtained representation. Finally, MU allows better use of previously unexplored regions, as expected from this kind of data augmentation. We show that our feature recombination helps the network better generalize and anticipate the coming of novel classes.

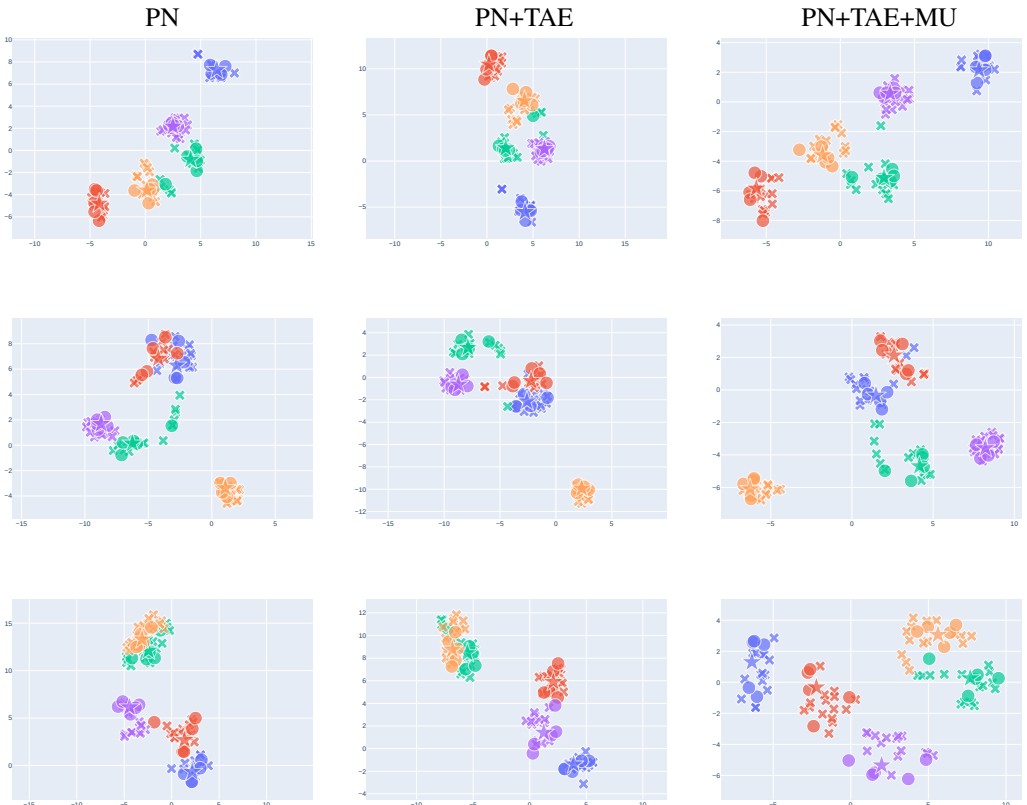

**Figure 4:** Visualization of latent spaces from the COIL-DEL dataset, through T-SNE dimensionality reduction. Each row is a different episode, the colors represent novel classes, the crosses are the queries, the circles are the supports and the stars are the prototypes. The left column is produced with the base model PN, the middle one with the PN+TAE model, the right one with the full model PN+TAE+MU. This comparison shows the TAE and MU regularizations improve the class separation in the latent space, with MU proving essential to obtain accurate latent clusters.

# 6   Conclusions

**Limitations.**   Employing a graph neural network embedder, the proposed approach may inherit known issues such as the presence of information bottlenecks [53] and over smoothing [13]. These may be aggravated by the additional aggregation required to compute the prototypes, as the readout function to obtain a graph-level representation is already an aggregation of the node embeddings. Also, the nearest-neighbour association in the final embedding assumes that it enjoys a euclidean metric. While this is an excellent local approximation, we expect it may lead to imprecision. To overcome this, further improvements can be inspired by the Computer Vision community [51].

**Future works.**   In future work, we aim to enrich the latent space defined by the architecture, for instance, forcing the class prototypes in each episode to be sampled from a learnable distribution rather than directly computed as the mean of the supports. Moreover, it may be worth introducing an attention layer to have supports (or prototypes, directly) affect each other directly and not implicitly, as it now happens with the task embedding module. We also believe data augmentation is a crucial technique for the future of this task: the capacity to meaningfully inflate the small available datasets may result in a significant performance improvement. In this regard, we plan to extensively test the existing graph data augmentation techniques in the few-shot scenario and build upon MixUp to exploit different mixing strategies, such as non-linear interpolation.

**Conclusions.**   In this paper, we tackle the problem of few-shot graph classification, an under-explored problem in the broader machine learning community. We provide a modular and extensible

codebase to facilitate practitioners in the field and set a stable ground for fair comparisons. The latter contains re-implementations of the most relevant baselines and state-of-the-art works, allowing us to provide an overview of the possible approaches. Our findings show that while there is no one-fits-all approach for all the datasets, the overall best results are obtained by using a distance metric learning baseline. We then suggest valuable additions to the architecture, adapting a task-adaptive embedding procedure and designing a novel online graph data augmentation technique. Lastly, we prove their benefits for the problem over several datasets. We hope this work to encourage a reconsideration of the effectiveness of distance metric learning when dealing with graph-structured data. In fact, we believe metric learning to be incredibly fit for dealing with graphs, considering that the latent spaces encoded by graph neural networks are known to capture both topological features and node signals effectively. Most importantly, we hope this work and its artifacts to facilitate practitioners in the field and to encourage new ones to approach it.

## Acknowledgments

This work is supported by the ERC Grant no.802554 (SPECGEO) and an Alexander von Humboldt Foundation Research Fellowship.

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

# A  Data statistics

We report in Table 3 general statistics of the datasets considered in this work.

# B  Additional details

## B.1  Evaluation setting

The models are trained in an episodic framework by considering $N$-way $K$-shot episodes with the same $N$ and $K$ considered for the novel classes at test time. We use for each dataset the same $N$ and $K$ proposed by the works in which they were introduced. In particular, $K = 5, 10$ for all the datasets, while the number of classes $N$ is reported in Table 4. The best model used for evaluation is picked by employing early stopping over the validation set. The latter is composed of a random $20\%$ subset of the base samples for datasets in $\mathcal{D}_A$ while it is composed of samples from a disjoint set of novel classes, different from the ones used for testing, for datasets in $\mathcal{D}_B$.

The epochs contain $2000$, $500$ and $1$ episodes for train, val and test respectively. Finally, the number of queries $Q$ is set to $15$ for each class and for each dataset. Each episode has therefore in total $N * Q$ queries. The number of episodes in a batch is set to $32$ for all the datasets except that for Reddit, for which is set to $8$.

|  | Dataset | avg # nodes | avg # edges | # samples | # samples / class | # classes | # base | # val | # novel |
|---|---|---|---|---|---|---|---|---|---|
| $\mathcal{D}_B$ | COIL-DEL | 21.54 | 54.24 | 3900 | 39 | 96 | 60 | 16 | 20 |
|  | Graph-R52 | 30.92 | 165.78 | 8214 | *unbalanced* | 28 | 18 | 5 | 5 |
| $\mathcal{D}_A$ | TRIANGLES | 20.85 | 35.5 | 2010 | 201 | 10 | 7 | 0 | 3 |
|  | ENZYMES | 32.63 | 62.14 | 600 | 100 | 6 | 4 | 0 | 2 |
|  | Letter_high | 4.67 | 4.5 | 2250 | 150 | 15 | 11 | 0 | 4 |
|  | Reddit-12K | 391.41 | 456.89 | 1111 | 101 | 11 | 7 | 0 | 4 |

**Table 3:** Statistics of all the considered datasets. These are grouped according to whether they encompass a disjoint set of classes to be used for validation. Graph-R52 is the only one with a skewed distribution of samples over its classes.

| | N | Train (base classes) | Validation | Test (novel classes) |
|---|---|---|---|---|
| Graph-R52 | 2 | {3, 4, 6, 7, 8, 9, 10, 12, 15, 18, 19, 21, 22, 23, 24, 25, 26, 27} | {2, 5, 11, 13, 14} | {0, 1, 16, 17, 20} |
| COIL-DEL | 5 | {0, 1, 2, 3, 4, 5, 6, 7, 8, 9, 10, 11, 12, 13, 14, 15, 16, 17, 18, 19, 20, 21, 22, 23, 24, 25, 26, 27, 28, 29, 30, 31, 32, 33, 34, 35, 36, 37, 38, 39, 40, 41, 42, 43, 44, 45, 46, 47, 48, 49, 50, 51, 52,53, 54, 55, 56, 57, 58, 59, 60, 61, 62, 63} | {64, 65, 66, 67, 68, 69, 70, 71, 72, 73, 74, 75, 76, 77, 78, 79} | {80, 81, 82, 83, 84, 85, 86 87, 88, 89, 90, 91, 92, 93, 94, 95, 96, 97, 98, 99} |
| ENZYMES | 2 | {1, 3, 5, 6} | * | {2, 4} |
| Letter-High | 4 | {1, 9, 10, 2, 0, 3, 14, 5, 12, 13, 7} | * | {4, 6, 11, 8} |
| Reddit | 4 | {1, 3, 5, 6, 7, 9, 11} | * | {2, 4, 8, 10} |
| TRIANGLES | 3 | {1, 3, 4, 6, 7, 8, 9} | * | {2, 5, 10} |

**Table 4:** Split between base and novel classes for each dataset, chosen to be the same as the competitors. Datasets marked with a (*) do not have a disjoint set of classes for validation, so the validation set is a disjoint subsample of samples from the base classes.

We follow the same base-novel splits used by GSM and AS-MAML. These are shown in Table 4. The model configurations are described in Table 5. Hyperparameter values for TRIANGLES and Letter-High were found via Bayesian parameter search, while those for Graph-R52, COIL-DEL, ENZYMES and Reddit were set to the same set of manually found values after having observed an overall small benefit in employing searched parameters. For the evaluation, we randomly sample 5000 episodes containing support and query samples from the novel classes. We then compute the accuracy over the query samples.

| | $\mathcal{D}_A$ | | | | $\mathcal{D}_B$ | |
|---|---|---|---|---|---|---|
| | ENZYMES | Letter-High | Reddit | TRIANGLES | COIL-DEL | Graph-R52 |
| LR | 1e-4 | 1e-2 | 1e-4 | 1e-3 | 1e-4 | 1e-4 |
| Scaling factor | 7.5 | 90.0 | 7.5 | 7.5 | 7.5 | 7.5 |
| $\gamma_0$ init. | 0.0 | 0.0 | 0.0 | 0.0 | 0.0 | 0.0 |
| $\beta_0$ init. | 1.0 | 5.0 | 1.0 | 1.0 | 1.0 | 1.0 |
| $\lambda_{\text{mixup}}$ | 0.1 | 0.1 | 0.1 | 0.6 | 0.1 | 0.1 |
| $\lambda_{\text{reg}}$ | 0.1 | 0.3 | 0.1 | 0.8 | 0.1 | 0.1 |
| Global Pooling | mean | sum | mean | mean | mean | mean |
| Embedding dim. | 64 | 32 | 64 | 64 | 64 | 64 |
| # convs | 2 | 3 | 2 | 2 | 2 | 2 |
| Dropout | 0.0 | 0.7 | 0.0 | 0.5 | 0.0 | 0.0 |
| # GIN MLP layers | 2 | 2 | 2 | 1 | 2 | 2 |

**Table 5:** Model hyperparameters for the various datasets.

In GSM, the reported standard deviation is computed among a different number of runs of the same pretrained model for different support and query sets. Since they do not employ an episodic framework neither for training and for evaluation, their setting is not directly comparable to ours and therefore led us to re-implement it. We used the same hyperparameters employed in the original manuscript for the datasets in $\mathcal{D}_A$. For the datasets in $\mathcal{D}_B$, over which the original model has never been employed, we chose the number of superclasses to match the increased number of classes in the latter datasets, choosing a value of 4 and 10 for Graph-R52 and COIL-DEL respectively. Furthermore, for the transfer learning baselines we use the same setting of our re-implementation of GSM, but we set repeat the fine-tuning phase of the supports 10 times.

For the graph kernel methods, we use the Grakel library [47]. A SVM is used as the classifier for all three approaches with the kernel sets to "precomputed" as the graph kernel methods pass to it the similarity matrix. We employ the default parameters for all the graph kernels for all the datasets, excluding Graphlet on R-52 and Reddit where we use a graphlet size equals to 3 instead of the default value 5, where the computational costs were infeasible due to the size of graphs.

Finally, since AS-MAML reports the 0.95 confidence interval, we also re-implement this work using the same hyperparameters of the original work, allowing us to retrieve the results on the remaining datasets.

## B.2 Episodes generation and training procedures

We outline in Algorithm 1 the pseudo-code to generate the $N$-way $K$-shot episodes. Algorithm 2 and Algorithm 3 then present the training pipeline for `ProtoNet` and `MAML` respectively.

---

**Algorithm 1** Episodes generation.

---

1: **procedure** GENERATE_EPISODES($\mathcal{G}$: dataset of graphs, $N_{\text{episodes}}$: int, $K$: int, $Q$: int)
2:  $C \leftarrow$ classes in $\mathcal{G}$
3:  $\mathcal{E} \leftarrow []$
4:  **for all** $i$ in $N_{\text{episodes}}$ **do**
5:    $e \leftarrow []$
6:    $C_{\text{episode}} \leftarrow$ sample $N$ classes from $C$
7:    **for all** $c$ in $C_{\text{episode}}$ **do**
8:      $\mathcal{S} \leftarrow$ sample $K$ graphs with class $c$
9:      $\mathcal{Q} \leftarrow$ sample $Q$ graphs with class $c$, $\mathcal{S} \cap \mathcal{Q} = \varnothing$
10:     $e \leftarrow (\mathcal{S}, \mathcal{Q})$
11:    **end for**
12:    $\mathcal{E} \leftarrow \mathcal{E} + e$
13:  **end for**
14:  **return** $\mathcal{E}$
15: **end procedure**

---

**Algorithm 2** Prototypical Networks training.

---

1: **procedure** TRAIN($\mathcal{E}$: dataset of episodes, $d$: distance function, $\mathcal{M}$: model)
2:  $\ell \leftarrow 0$
3:  **for all** $e$ in $\mathcal{E}$ **do**
4:    $(\mathcal{S}, \mathcal{Q}) \leftarrow e$
5:    $\bar{\mathcal{S}} \leftarrow \mathcal{M}(\mathcal{S})$                      $\triangleright$ embed supports
6:    $\bar{\mathcal{Q}} \leftarrow \mathcal{M}(\mathcal{Q})$                      $\triangleright$ embed queries
7:    $\mathcal{P} \leftarrow []$
8:    **for all** $c$ in $C_{\text{episode}}$ **do**              $\triangleright$ classes of the episode
9:      $\bar{\mathcal{S}}_c \leftarrow$ supports with class $c$
10:     $p_c \leftarrow$ mean $(\bar{\mathcal{S}}_c)$
11:     $\mathcal{P} \leftarrow \mathcal{P} + p_c$
12:    **end for**
13:    $\mathbf{D} \leftarrow$ matrix $\in \mathbb{R}^{Q \times N}$, $D_{ij} = d(\bar{\mathcal{Q}}_i, \mathcal{P}_j)$
14:    $\ell \leftarrow \ell + \text{CrossEntropy}(-\mathbf{D}, \mathbf{Y}_{\mathcal{Q}})$        $\triangleright$ $\mathbf{Y}_{\mathcal{Q}}$ ground truth
15:  **end for**
16:  $\mathcal{M} \leftarrow \text{SGD}(\mathcal{M}, \ell)$
17: **end procedure**

---

## B.3 Efficiency analysis

Table 6 reports the training time and number of episodes of our approach over each dataset. Table 7 instead shows how the model compares in training and inference times with respect to the other considered models over `Graph-R52`.

| | $\mathcal{D}_{\text{A}}$ | | | | | | | | $\mathcal{D}_{\text{B}}$ | | | |
| --- | --- | --- | --- | --- | --- | --- | --- | --- | --- | --- | --- | --- |
| | ENZYMES | | Letter-High | | Reddit | | TRIANGLES | | COIL-DEL | | Graph-R52 | |
| | 5-shot | 10-shot | 5-shot | 10-shot | 5-shot | 10-shot | 5-shot | 10-shot | 5-shot | 10-shot | 5-shot | 10-shot |
| Time (seconds) | 1058 | 817 | 8493 | 3698 | 1846 | 2156 | 1600 | 1252 | 4269 | 5948 | 1449 | 1388 |
| Episodes | 192 | 192 | 8320 | 1792 | 128 | 64 | 4608 | 3072 | 1856 | 4544 | 1920 | 1536 |

**Table 6:** Training time in seconds and number of episodes over the various datasets with varying number of shots $k$. These include the whole training time with early stopping enabled. All the computation was carried on a NVIDIA 2080Ti GPU with an Intel(R) Core(TM) i7-9700K CPU.

---

**Algorithm 3** Meta Learning pipeline.

1: **procedure** TRAIN($\mathcal{E}$: dataset of episodes, $N_{in}$: number of inner steps, $\mathcal{M}$: model)
2:     $\ell_{out} \leftarrow 0$
3:     **for all** $e$ in $\mathcal{E}$ **do**                                                   ▷ outer loop
4:         $(\mathcal{S}, \mathcal{Q}) \leftarrow e$
5:         $\mathcal{M}' \leftarrow \text{copy}(\mathcal{M})$
6:         **for all** $i$ in $N_{in}$ **do**                                         ▷ inner loop
7:             $\hat{\mathbf{Y}}_\mathcal{S} \leftarrow \mathcal{M}'(\mathcal{S})$
8:             $\ell_{in} \leftarrow \text{CrossEntropy}(\hat{\mathbf{Y}}_\mathcal{S}, \mathbf{Y}_\mathcal{S})$             ▷ $\mathbf{Y}_\mathcal{S}$ ground truth
9:             $\mathcal{M}' \leftarrow \text{SGD}(\mathcal{M}', \ell_{in})$
10:         **end for**
11:         $\hat{\mathbf{Y}}_\mathcal{Q} \leftarrow \mathcal{M}(\mathcal{Q})$
12:         $\ell_{out} \leftarrow \ell_{out} + \text{CrossEntropy}(\hat{\mathbf{Y}}_\mathcal{Q}, \mathbf{Y}_\mathcal{Q})$         ▷ $\mathbf{Y}_\mathcal{Q}$ ground truth
13:     **end for**
14:     $\mathcal{M} \leftarrow \text{SGD}(\mathcal{M}, \ell_{out})$
15: **end procedure**

---

|  | GSM* | | MAML | | PN | | PN+TAE | | PN+TAE+MU | |
|---|---|---|---|---|---|---|---|---|---|---|
|  | 5-shot | 10-shot | 5-shot | 10-shot | 5-shot | 10-shot | 5-shot | 10-shot | 5-shot | 10-shot |
| Training time | 0:50:03 | 0:56:03 | 0:32:57 | 0:32:28 | 0:12:07 | 0:19:11 | 0:16:21 | 0:25:15 | 0:24:09 | 0:23:08 |
| Inference time | 2.82s | 3.18s | 0.05s | 0.05s | 0.05s | 0.07s | 0.05s | 0.06s | 0.06s | 0.06s |

**Table 7:** Training and inference times of the considered models.

### B.4 Difference with SMF-GIN

The main difference of our `ProtoNet` baseline and the architecture proposed by `SMF-GIN` [30] lies in the loss computation, as in `SMF-GIN` the cross-entropy is computed over the one-hot prediction for the query and the ground truth label. Differently, we instead directly compute the cross-entropy between the predicted class probability vector and the ground truth label vector, the first obtained as the softmax over the additive inverse of the query-prototypes distances. By doing so, we preserve the quantitative distance information for all the classes, which is discarded if only the one-hot vector prediction is considered. The superior performance can be appreciated in the results for the only common benchmark that is considered in `SMF-GIN`, i.e. `TRIANGLES`, where our our `ProtoNet` baseline achieves an accuracy of $86.64$ versus the $79.8$ reported by `SMF-GIN`. The latter result empirically confirms the importance of faithfully adhering to the original `ProtoNet` pipeline.

## C Qualitative Analysis

More insight into the learned latent space is provided in Figures 5 to 7. In Figure 5, the latent space of different episodes for the `Graph-R52` dataset is shown considering the three presented models. It is worth noting that, on the `Graph-R52` dataset, the PN+TAE model creates better clusters than the PN model, and these are slightly improved with the addition of MU. Nevertheless, the benefits of adding MU are not as clearly visible as they are for `COIL-DEL`, and this is also reflected in the less prominent benefit in accuracy. Subsequently, in Figure 6 we present the latent space of a novel episode produced by the datasets belonging to $\mathcal{D}_\text{A}$, namely `ENZYMES`, `Letter-High`, `Reddit` and `TRIANGLES`. We compare the T-SNE obtained by our full model with the one obtained by GSM* (our re-implementation of GSM). As can be seen, our model is more successful at separating samples into clusters than GSM*. Finally, in Figure 7 we show the latent space of a novel episode produced by the datasets belonging to $\mathcal{D}_\text{B}$. As before, the T-SNE plot demonstrates the better separation ability of our full model than GSM* also for these datasets.

### C.1 Standard deviation-aware global pooling

As it is typical in graph representation learning, graph-level embeddings are obtained in this work by aggregating the node embeddings with some permutation invariant function, such as the mean or the

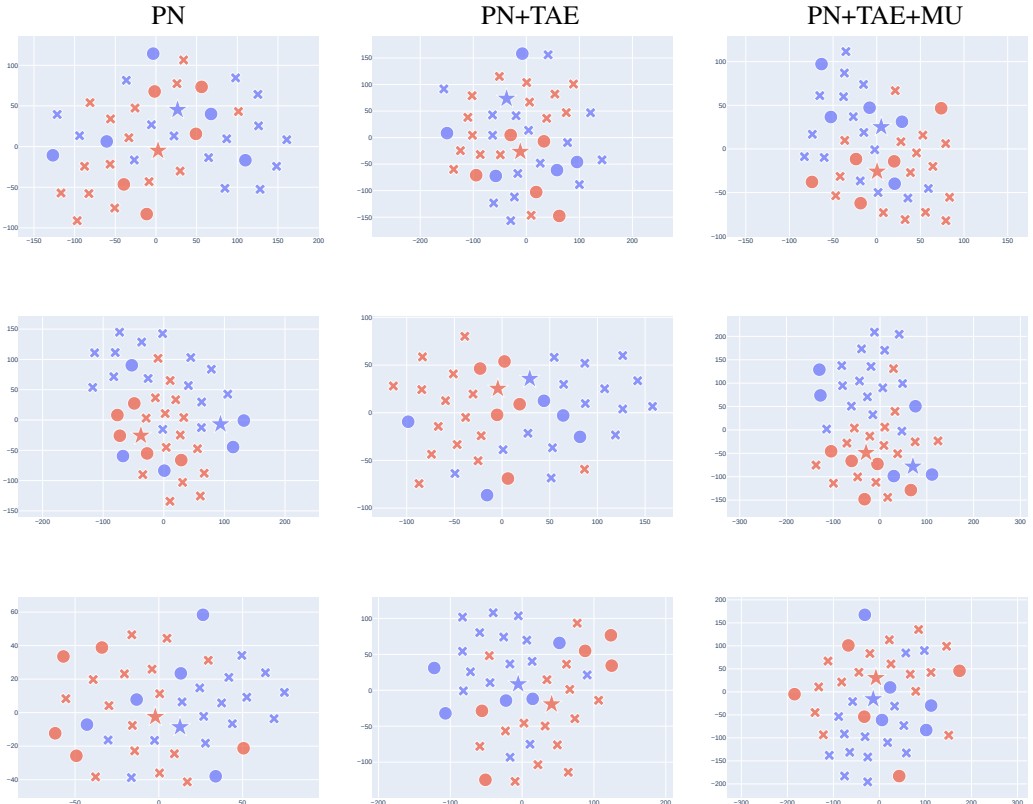

**Figure 5:** Visualization of novel episodes' latent spaces from the `Graph-R52` dataset, through T-SNE dimensionality reduction. Each row is a different episode, the colors represent novel classes, the crosses are the queries, the circles are the supports and the stars are the prototypes. The left column is produced with the base model PN, the middle one with the PN+TAE model, the right one with the full model PN+TAE+MU. This comparison shows that the TAE and MU regularizations improve the class separation in the latent space, although less remarkably than in `COIL-DEL`.

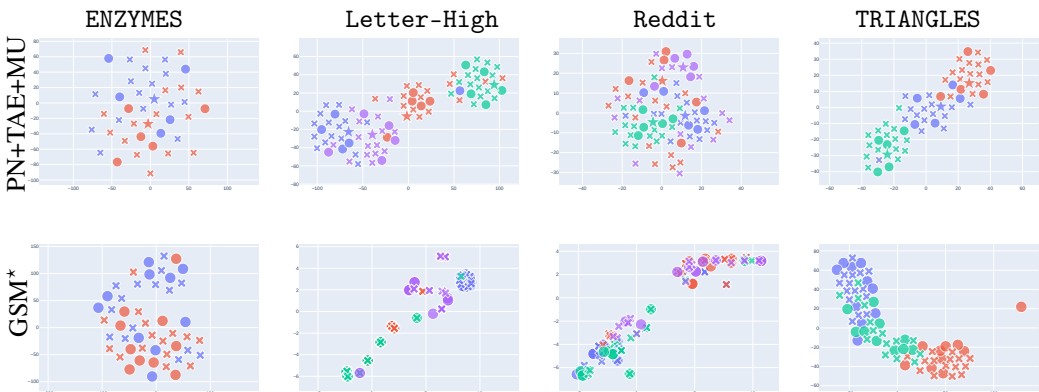

**Figure 6:** T-SNE visualization of a novel episode's latent space from the datasets belonging to $\mathcal{D}_{\mathrm{A}}$. The first row shows the T-SNE produced with our full model (PN+TAE+MU), while the second one shows the plots produced with GSM$^{\star}$. In each plot, the colors represent novel classes, the crosses are the queries and the circles are the supports. In addition, since our model works with prototypes, these are represented by the stars only in the plots of the first row.

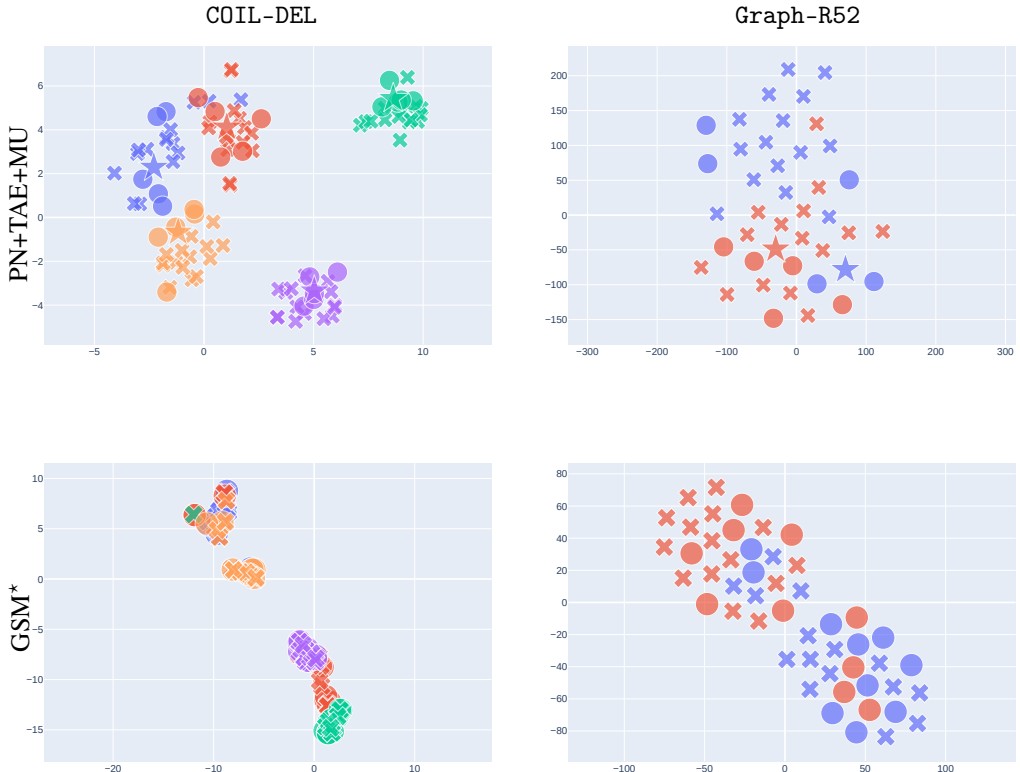

**Figure 7:** T-SNE visualization of a novel episode's latent space from the datasets belonging to $\mathcal{D}_B$. The first row shows the T-SNE produced with our full model (PN+TAE+MU), while the second one shows the plots produced with GSM$^\star$. In each plot, the colors represent novel classes, the crosses are the queries and the circles are the supports. In addition, since our model works with prototypes, these are represented by the stars only in the plots of the first row.

| Pooling | Graph-R52 | | | | COIL-DEL | | | |
|---|---|---|---|---|---|---|---|---|
| | 5-shot | | 10-shot | | 5-shot | | 10-shot | |
| mean | **77.9** | $\pm 11.8$ | **81.5** | $\pm 10.4$ | **87.7** | $\pm 9.2$ | **90.5** | $\pm 7.7$ |
| mean + var | 74.41 | $\pm 12.67$ | 79.45 | $\pm 10.12$ | 86.45 | $\pm 10.19$ | 88.78 | $\pm 8.99$ |

**Table 8:** Macro accuracy scores for mean var pooling versus standard mean pooling.

sum. As a prototype is already defined as the mean of the samples for the corresponding class, the risk of obtaining over-smoothed representations increases. Aiming to alleviate this issue, we also experiment with graph-level embeddings containing information about both the mean of the node embeddings as well as the standard deviation. In particular, we first halve the dimension of each node embedding with a learnable linear transformation, and then compute mean and standard deviation of the transformed embeddings. The final graph-level embedding will be the concatenation of the mean and standard deviation of its node embeddings. The model employing this variant of pooling is called 'mean + var' in Table 8. Nevertheless, we observe on-par or slightly worse results in accuracy when employing this variant. Additional tuning may be required to take full advantage of this information, leaving an interesting future direction to investigate.

| Model | Graph-R52 | | | | COIL-DEL | | | | mean | |
|---|---|---|---|---|---|---|---|---|---|---|
| | 5-shot | | 10-shot | | 5-shot | | 10-shot | | 5-shot | 10-shot |
| PN | 73.1 | ±12.1 | 78.0 | ±10.6 | 85.5 | ±9.8 | 87.2 | ±9.3 | 79.3 | 82.6 |
| PN+MU | 73.49 | ±12.39 | 78.25 | ±11.04 | 85.41 | ±10.1 | 87.65 | ±9.21 | 79.45 | 82.95 |
| PN+TAE | 77.9 | ±11.8 | 81.3 | ±10.6 | 86.4 | ±9.6 | 88.8 | ±8.5 | 82.1 | 85.0 |
| PN+TAE+MU | 77.9 | ±11.8 | 81.5 | ±10.4 | **87.7** | ±9.2 | **90.5** | ±7.7 | **82.8** | **86.0** |

**Table 9:** Ablation study over different *k-shot* settings.

## C.2 Ablation study

We report here the results of the ablation study over Graph-R52 and COIL-DEL. As it is evident from the table, MixUp alone does not yield a significant boost in accuracy, while providing a more sensible increment when coupled with Task Adaptive Embeddings. The latter allows samples to be embedded in the most convenient way for the episode at hand, possibly also enabling more meaningful mixed samples. We note, however, that the MixUp configuration was evaluated with the same hyperparameters used in the full model, and hence the actual results may be slightly better.

## C.3 MixUp and class similarities

In this section, we investigate the effect of MixUp on the similarity among different classes. To this end, we compute the mean of 100 random samples for each class, obtaining a representative for each class, and compute the similarity among all possible pairs of class representatives. The similarity is based on the squared L2 distance which is used during the optimization. We run the same computation for a model trained with MixUp and one without. In order to have a more immediately understandable visualization, we compute for each class its mean similarity with the other classes, which is basically the mean over the column dimension of the similarity matrix. We then compute the difference of these values between vanilla and MixUp, getting the vectors in Figure 8. It is immediate to see that the vectors contain all positive values, indicating that the classes are actually more different when employing MixUp. This observation is coherent with the improved classification scores, as it is particularly crucial for a metric-based model to have an embedding space in which classes are easily discriminable using the metric that is used in the optimization.

## C.4 Class imbalance

We believe class imbalance to be under-investigated in episodic frameworks. In our case, the mean of the supports to create the prototype is still going to be computed over the same fixed number of samples ($K$) for each episode. While this avoids cases in which a class prototype is computed over a large number of samples and one is computed over just a few, it is not immediately clear how much effect data imbalance may have in such a scenario. In general, it is intuitive to assume that the model will learn a more suitable representation for data-abundant classes than for the rarer ones. To see the effect of data imbalance on our model, we also evaluated on the imbalanced dataset Graph-R52. As can be seen in Figure 9, the dataset in fact exhibits a severely skewed sample distribution among the classes. The lesser improvement compared to what we gain on other datasets may suggest that our model may be hindered by class imbalance. However, this behavior might be inherited from the episodic setting itself, as it has been speculated to yield worse results when dealing with imbalanced datasets [16]. We, therefore, aim to replace the random sample selection in the episode generation with an active one, as this has been shown to be particularly beneficial for class-imbalanced tasks [16]. This extension is left for future work.

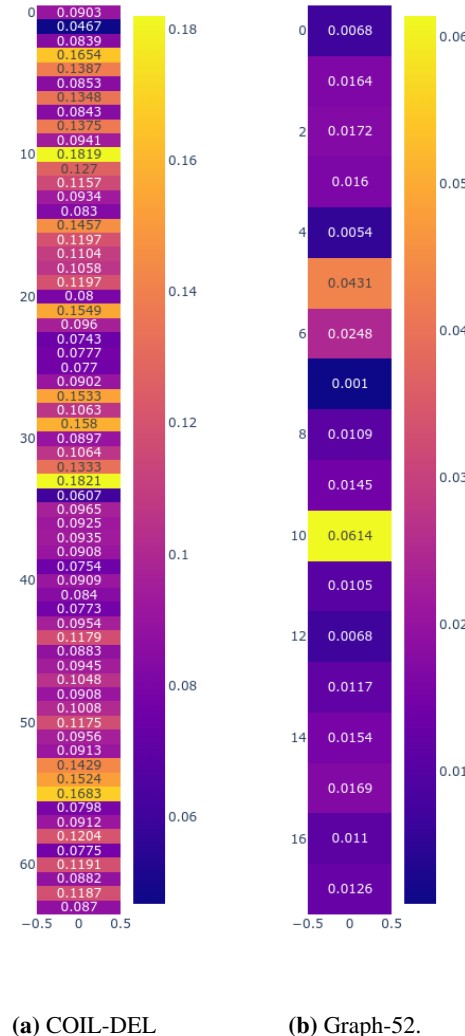

**(a)** COIL-DEL       **(b)** Graph-52.

**Figure 8:** Difference in mean class similarity between `PN+TAE` and `PN+TAE+MU`.

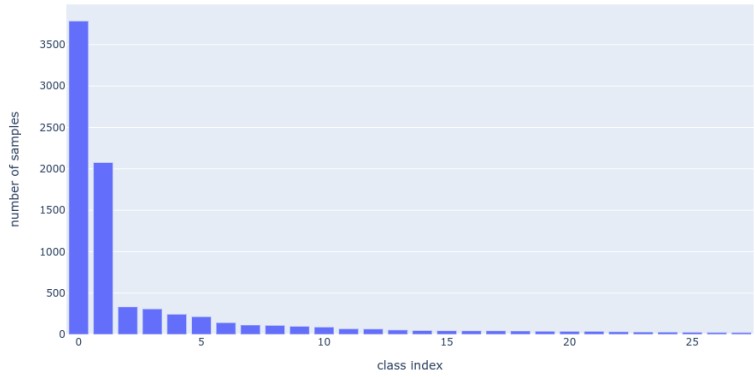

**Figure 9:** Sample distribution for `Graph-R52`.

