# OpenReview forum: "Metric Based Few-Shot Graph Classification"
_logconference.io/LOG/2022/Conference — LoG 2022 Poster_

### Official Review · Reviewer_6W4U · 2022-10-18

**Overall Score:** 6
**Confidence:** 4

**Review:**

This work tackles the problem of few-shot graph classification. Specifically, the authors first provide a codebase by re-implementing the SOTA methods. Then, they propose two additions to the model architecture, including a task-adaptive embedding procedure and an online graph data augmentation technique. Finally, they prove the effectiveness of these modules on benchmark datasets.

Strengths:
* The codebase for these experiments is provided, which is comprehensive and usable.
* The authors provide a relatively fair benchmark to help researchers better understand the strengths and weaknesses of SOTA methods.
* Qualitative analysis makes a lot of sense to me.

Weaknesses:
* It seems that the model has not been tuned extensively. The hyperparameters are provided in the appendix, but it is not sure how the Authors came up with these numbers. Are they properly tuned, reused from the literature, or hand-picked from multiple experimental runs?
* Can the two proposed modules be merged with modules other than PN? Also, the results of these two modules do not seem to be very promising and the authors need to add extra ablation experiments to prove the effectiveness of the modules

Questions:
* It would be interesting to see a more detailed tuning scheme.
* The extensibility of the codebase should be discussed.

---

### Official Review · Reviewer_v1cH · 2022-10-21

**Overall Score:** 6
**Confidence:** 4

**Review:**

This paper considers the task of few-shot graph classification, which has attracted some attention in recent years.  It benchmarks prototypical networks (PN) using a GNN feature extractor, showing that with two simple techniques--modulating the embeddings with a task embedding network (TAE), and mixup-style augmentation (MU)--it is competitive or superior to existing techniques.

The paper is well-organized and easy to follow.  Methodologically, the contributions are on the lighter side.  The main method is based on using PN in fact, PN has been used for graphs before (as the authors note when discussing the baseline method SMF-GIN), and by comparison the authors actually claim to depart less from the original PN setup.  The proposed techniques to improve performance are fairly simple extensions of standard techniques used in few-shot learning.  The main contribution of this work seems to lie not so much in the extent of the technical machinery it introduces to solve the problem of few-shot graph classification, but rather in showing that the state-of-the-art is at such a level that extensive technical machinery is not needed to yield top results.  For a field that is starting to develop, this can be a valuable contribution, giving additional papers competitive yet easily-implementable baselines to beat and helping the research community avoid putting the cart before the horse (developing complex methods that are not necessarily superior to simple baselines).

If the methodology is to be made a contribution, it should be contrasted more precisely with the very related existing works.  For example, the authors note that SMF-GIN also uses PN, but claims several times that their implementation is "a more faithful formulation of PN that shows far superior performance."  In what way are the design choices more faithful to the original paper, and why would that be expected to show superior performance? It is also claimed that the proposed mixup variation, in contrast to other methods that perform mixup of graph data in the latent space, "is tailored for distance metric learning, making full use of the similarity among samples and class prototypes." It would be helpful to clarify how the proposed MU method makes full use of the class prototypes or why that is expected to be helpful.  It is interesting that the proposed MU method mixes up class probability distributions (which are obtained by comparing to prototypes) -- is that what is meant?  If so, maybe it could be clarified in the methodology section what benefit this provides for metric learning.  Finally, it would be helpful to perform the quasi-ablation study (considering PN without TAE and/or MU) on more datasets than just the two in $\mathcal{D}_B$, to show to what extent a simple PN is sufficient for the task and what extent TAE and/or MU, introduced here, are necessary.

Experiments are broken up into two sets of datasets: $\mathcal{D}_A$, as used in the paper introducing the baseline GSM, and $\mathcal{D}_B$, as also used in the paper introducing the baseline AS-MAML.  Again, the authors note that the datasets in $\mathcal{D}_A$ are not big enough to have a disjoint set of validation classes, and correctly point out another limitation with these existing benchmarks, namely that this makes it harder to have a validation criterion that is an accurate proximity for the few-shot performance in unseen classes.  Even $\mathcal{D}_B$ is a comparatively small benchmark.  It may be worth evaluating on a larger benchmark from an real-world application, such as FS-Mol [1].  Note that prototypical networks were benchmarked on this dataset (to be fair, also using SMILES strings and other domain knowledge), and perhaps corroborating the findings of the paper, found to yield superior performance compared to other meta-learning paradigms such as MAML. Even benchmarking PN + the TAE + MU on just the graph portion of this data might be a more informative benchmark than either $\mathcal{D}_A$ or $\mathcal{D}_B$.

[1] Stanley et al. FS-Mol: A Few-Shot Learning Dataset of Molecules. NeurIPS Datasets and Benchmarks, 2021.

---

### Official Review · Reviewer_yWyD · 2022-10-22

**Overall Score:** 5
**Confidence:** 4

**Review:**

[Summarization]
In this paper, the authors study the graph few-shot classification problem and propose a straightforward solution by combining off-the-shelf solutions.

[Strong points]
The experimental results seem promising.
The literature review is complete.

[Weak points]
The contribution is incremental and limits novelty.
Inconsistent symbols. In Equations (1) and (4), what is the difference between h^{(l)} and h_{l}?
The episodic training pseudo-code can be added to help the systematical illustration for training procedures.

[Questions]
From Table 2, why did +MU not improve the performance obviously?

[Improvements]
The reasoning behind why this specific combination is not fully discussed. The authors can discuss why this combination is indispensable, and why other off-the-shelf meta-learning diagrams and specific methods not introduced are not helpful.

---

### Official Review · Reviewer_RR11 · 2022-10-27

**Overall Score:** 6
**Confidence:** 3

**Review:**

### 1. Summary
Implementation of many standard techniques for few-shot learning, but adapted to graphs, and the development of their own model, based on learning a sort of embedding interpolation between classes

### 2. Strong / weak points

**Strong**
- Paper is relatively well written and illustrated, easy to understand
- Results are good across a wide set of benchmarks
- Method is simple, yet novel

**Weak**
- No clear study of why and how the model works, weak theoretically

### 3. Recommendation
6 - Accept.
3 - Somewhat Confident

### 4. Arguments
The novelty and simplicity factors, together, makes for a strong and simple paper. The framework allows new researcher to build their ideas easily. The paper is well written.

Main issue is that their model lacks theoretical motivation and study, and many questions are left unanswered.

Willing to raise my score.

### 5. Questions

- It seems to me that the way the mask is made, it will force the model to learn a very similar embedding across classes since any variation of embedding will be considered as an interpolation between classes. Does that kind of behavior appear? Will it likely help? Is the proposed model a form of regularization?
- How to adapt the model for regression?
- Is the mean of the distribution sufficient? Why not use a standard deviation as well to better interpolate? It seems to me that, when datasets are large, then the mean of the embedding is no longer representative? Of course, if my first question is right, then standard deviation should be low.
- What happens in case of imbalance? I assume the mean becomes even less relevant and it becomes harder to interpolate between classes? Would you be willing to do such experiment by artificially creating imbalance?

### 7. Paper formatting
Looks good

---

### Meta-Review · Area_Chair_UanW · 2022-11-24

**Confidence:** 5
**Recommendation:** Borderline and needs further discussi…

**Meta Review:**

This paper proposes a new approach for few-shot graph classification. Though this paper is clearly written and easy to read. I do think this method just combines several existing approaches that have already been used in graph few-shot learning. For example, task-adaptive embedding has already been used in [69] in 2020 and mixup is a very commonly used trick. Thus, I think this is a borderline paper (tend to reject it) and needs more discussion.

---

### Decision · Program_Chairs · 2022-11-23

**Decision:**

Accept (Poster)

**Comment:**

Reviewers were somewhat undecided about this paper but raised good suggestions for improvement. We encourage the authors to incorporate such improvements in their camera-ready version.